# Light-Induced Changes in Secondary Metabolite Production of *Trichoderma atroviride*

**DOI:** 10.3390/jof9080785

**Published:** 2023-07-26

**Authors:** Kristina Missbach, Daniel Flatschacher, Christoph Bueschl, Jonathan Matthew Samson, Stefan Leibetseder, Martina Marchetti-Deschmann, Susanne Zeilinger, Rainer Schuhmacher

**Affiliations:** 1Department of Agrobiotechnology IFA-Tulln, Institute of Bioanalytics and Agro-Metabolomics, University of Natural Resources and Life Sciences Vienna (BOKU), 3430 Tulln, Austria; kristina.missbach@boku.ac.at (K.M.);; 2Department of Microbiology, Universität Innsbruck, 6020 Innsbruck, Austria; 3Institute of Chemical Technologies and Analytics, TU Wien, 1060 Vienna, Austria; stefan.kirnbauer@tuwien.ac.at (S.L.);

**Keywords:** *Trichoderma atroviride*, secondary metabolome, light response

## Abstract

Many studies aim at maximizing fungal secondary metabolite production but the influence of light during cultivation has often been neglected. Here, we combined an untargeted isotope-assisted liquid chromatography–high-resolution mass spectrometry-based metabolomics approach with standardized cultivation of *Trichoderma atroviride* under three defined light regimes (darkness (PD), reduced light (RL) exposure, and 12/12 h light/dark cycle (LD)) to systematically determine the effect of light on secondary metabolite production. Comparative analyses revealed a similar metabolite profile upon cultivation in PD and RL, whereas LD treatment had an inhibiting effect on both the number and abundance of metabolites. Additionally, the spatial distribution of the detected metabolites for PD and RL was analyzed. From the more than 500 detected metabolites, only 25 were exclusively produced upon fungal growth in darkness and 85 were significantly more abundant in darkness. The majority were detected under both cultivation conditions and annotation revealed a cluster of substances whose production followed the pattern observed for the well-known *T. atroviride* metabolite 6-pentyl-alpha-pyrone. We conclude that cultivation of *T. atroviride* under RL can be used to maximize secondary metabolite production.

## 1. Introduction

*Trichoderma* spp. are soil-borne filamentous fungi, widely used in biological control against plant pathogens [1,2]. Their broad application can be attributed to their multiple modes of action including metabolic diversity [3]. The antagonistic activity of *Trichoderma* species includes nutrient competition, mycoparasitism, the biosynthesis of hydrolytic enzymes, and the production of bioactive secondary metabolites. The current knowledge of the role of *Trichoderma* secondary metabolites has been summarized in [4,5,6,7]. 

Analyses of the genomes of *Trichoderma* spp. showed that there is significantly more potential for the production of secondary metabolites than previously assumed [8,9]. The majority of the secondary metabolites in fungi are known to be derived from transcriptionally co-regulated genes organized as clusters (biosynthetic gene clusters or BCGs) [10,11]. While fungal genomes harbor a large potential for secondary metabolite production, many have yet to be detected and identified, since the fungus does not produce them under standard laboratory conditions [12,13]. Moreover, many low molecular weight natural products are only produced in small amounts, in complex matrices, or when the fungus is triggered by certain cues. Advanced dereplication strategies to identify known compounds and advanced OMICS strategies are needed to find novel metabolites of interest [14,15].

Fungi are a highly diverse group of organisms with a complex and dynamic taxonomy [16,17]. This can make it difficult to assign specific secondary metabolites to a particular species. Additionally, the production of secondary metabolites is highly strain specific and can vary depending on environmental conditions and the presence of other organisms [18,19]. This complexity makes it challenging to predict which compounds are produced by a particular species or strain. For example, the Antibase 2017 database [20] contains only 14 metabolites of *Trichoderma atroviride*. Nevertheless, due to changes in fungal taxonomy, it cannot be ruled out that of the 482 entries for *Trichoderma* spp., some can also be produced by *Trichoderma atroviride* [20].

In the formation of secondary metabolites by filamentous fungi, various metabolic pathways can be distinguished, which also give rise to the names of the synthesized substances. For *T. atroviride,* those pathways include: 

Polyketides, a structurally diverse group of substances that are mostly synthesized from acetyl-CoA and malonyl-CoA by polyketide synthases [19]. While 10-25 PKS genes have been found in the *Trichoderma* genomes, their metabolic products are still unknown [8,9]. 

Terpenes are produced by terpene synthases from multiple activated forms of isoprene (C_5_H_8_). Depending on the number of isoprene units, monoterpenes, sesquiterpenes, etc., are formed [21,22] and, due to highly reactive metabolic intermediates, terpene synthases typically generate a mixture of closely related compounds rather than a single terpene [8,23]. Discovered sesquiterpenoids produced by *T. atroviride* include trichoderiol A-B, but genome analysis revealed the potential to produce more yet unknown terpenoids [12,21,24].

Non-ribosomal peptides are produced by large multi-modular enzyme complexes and typically contain 2-amino-isobutyric acid (Aib). Moreover, most of them share an acetylated N-terminus and an amino alcohol at the C terminus [25]. Peptaibols are known for their “microheterogeneity”, referring to their production as mixtures of structurally closely related peptides [26]. Peptaibols produced by *T. atroviride* include atroviridins A-C, trichorzianines, and trichoatrokontins [27,28]. 

An interesting property shared by numerous fungal metabolites is a high vapor pressure at room temperature. They can be both metabolic intermediates as well as end products of numerous biosynthetic routes and chemical structure classes. They include 6-pentyl-α-pyrone (6-PP), the most widely studied secondary metabolite of *T. atroviride* [29]. 6-PP was discovered in 1972 [30], and since then, it has been investigated thoroughly, although the biosynthetic route still remains to be identified [31,32,33].

Light, injury, and co-cultivation with fungal plant pathogens, amongst other factors, have been reported to affect the production of secondary metabolites in *T. atroviride* [18,32,34,35,36,37,38]. Although the influence of light has long been neglected, even short light pulses are able to trigger the expression of certain genes and to generate biochemical and developmental changes resulting in, e.g., the formation of conidia [39,40,41,42].

In this study, we have systematically investigated the effect of three defined light conditions on the production of secondary metabolites secreted by *T. atroviride*. We used an untargeted, stable isotope-assisted metabolomics approach [43] to find as many known as well as unknown true fungal substances as possible. Cultivation under complete darkness or reduced light turned out to favor secondary metabolite biosynthesis, while white light had an inhibitory effect. The favored light conditions were then retested, also investigating the spatial distribution of the produced metabolites in the fungal culture. 

## 2. Materials and Methods

### 2.1. Cultivation and Growth Conditions

*T. atroviride* strain IMI206040 (ATCC 204676) was used in this study. Cultures were grown and maintained on potato dextrose agar (PDA; Becton Dickinson, Le Pont-de-Claix, France).

#### 2.1.1. Cultivation on Solid Minimal Medium

For conditioning the fungus for metabolomics experiments, a 6 mm diameter mycelia-covered agar plug of the actively growing colony margin was transferred to the center of a fresh petri dish (94 mm × 16 mm, Greiner Bio-One GmbH, Kremsmünster, Austria) containing minimal medium (0.52 g L^−1^ KCl (Roth); 0.52 g L^−1^ MgSO4 × 7(H_2_O) (Roth); 1.52 g L^−1^ KH_2_PO_4_ (Fluka) 0.0004 g L^−1^ Na_2_B_4_O_7_ × 10(H_2_O) (Roth); 0.004 g L^−1^ CuSO_4_ × 5(H_2_O) (Sigma-Aldrich, Burlington, MA, USA); 0.008 g L^−1^ FeSO_4_ × 7(H_2_O) (Roth); 0.008 g L^−1^ MnSO_4_ × 4(H_2_O) (Sigma-Aldrich, Burlington, MA, USA); 0.005 g L^−1^ Na_2_MoO_4_ × 7(H_2_O) Sigma-Aldrich, Burlington, MA, USA); 0.08 g L^−1^ ZnSO_4_ × 7(H_2_O) (Roth); 2.64 g L^−1^ (NH_4_)_2_SO_4_ (Roth); 1.6% Agarose (NEEO ultra-quality; Roth, Karlsruhe, Germany) pH 5.5). Moreover, 10 g L^−1^ native or labeled glucose was used as a sole carbon source. For cultivation on native (^12^C) glucose, D (+)-glucose (≥99.5% purity; Sigma-Aldrich, Burlington, MA, USA) was used. For the global labeling of the metabolome, the fungus was cultivated on ^13^C_6_ glucose (U-^13^C_6_ D (+)-glucose; 99.9%; Cambridge Isotope Laboratories, Inc., Woburn, MA, USA). Cultures were incubated for 3 days at 25 °C.

#### 2.1.2. Cultivation in a Miniaturized Growth Assay

The minimal medium was cast directly onto microscope slides (Thermo Fisher Scientific, San Jose, CA, USA) using a custom-made casting chamber, resulting in a 4 mm thick layer of solid growth medium. After inoculation close to the left edge of the slide, cultures were incubated at 25 °C and 70% humidity in an incubator (Konstantklima-Kammer HPP 260- Memmert, Schwabach, Germany) with the following light regimes: (A) White light/dark cycle (12:12 h; 8000 Lux) (LD), (B) permanent darkness (PD), or (C) reduced light exposure (daily exposure to daylight for 10 min in total) (RL). The cultures were grown until the fungal colony reached the middle of the glass slide (approx. 72 h).

### 2.2. Sample Preparation

Immediately prior to sampling, the hyphal growth front was recorded and the culture slide was photographed. An exemplary picture is shown in Figure 1; all photos are provided in Appendix A. For the sampled cultures grown in darkness, photographing was omitted to avoid any light exposure. 

#### 2.2.1. Comparison of the Three Light Conditions

For each light condition, five replicates with native glucose and one slide with labeled glucose were cultivated. The solid medium on the culture slides was cut at 1 mm from the hyphal growth front and at a 90° angle to the growth direction. The overgrown section of the medium was harvested (red in Figure 1). The agarose plug for inoculation was removed and discarded before the samples were frozen in liquid nitrogen.

#### 2.2.2. Analysis of Spatial Metabolite Distribution

Culture slides were divided into five equally sized sections with a width of 1 cm each, starting next to, but not including the inoculation plug. Each section was then harvested and frozen separately.

#### 2.2.3. Sample Extraction

Each frozen sample was extracted with 5× its weight in acetonitrile (*m*/*v*) (ACN) (Riedel de Haen, Honeywell, LC-MS grade > 99.9% purity, Seelze, Germany): water (1:1, *v*/*v*) containing 0.1% formic acid (FA), vortexed and ultrasonicated for 15 min. 

#### 2.2.4. Internal Standardization with the Pooled Labeled Extract

Extracts from the ^13^C-labeled cultures were pooled and prepared in the same way as the native extracts. Then, each native sample extract was spiked with the same amount of labeled global pool extract. Spiked extracts were transferred into HPLC vials and immediately measured with LC-HRMS.

### 2.3. LC-HRMS/MS Measurements

For all LC-HRMS measurements, a UHPLC system (Vanquish—Thermo Fisher Scientific, San Jose, CA, USA) coupled to an Orbitrap HESI-Q Exactive HF (Thermo Fisher Scientific, San Jose, CA, USA) was used. 

#### 2.3.1. Determination of Glucose Concentration

Fifty µL aliquots of the extracts obtained from the cultivations under the three light conditions (after step 2.2.3.) were diluted 1:20 (v:v) with H_2_O, transferred into HPLC vials, and the glucose concentration in the media was determined by hydrophilic interaction chromatography (HILIC). A ZIC^®^-pHILIC column (100 × 2.1 mm i.d., 5 µm particle size—Merck Millipore, Burlington, MA, USA) was used. The column temperature and eluent pre-heater were set to 35 °C, a sample injection volume of 1 µL and a constant flow rate of 300 μL·min−1 were set. The eluents were 10 mM aqueous ammonium formate (Merk, Darmstadt, Germany) with pH 6 (Eluent A) and ACN (HiPerSolv Chromanorm, HPLC gradient grade, VWR, Vienna, Austria) containing 5% aqueous ammonium formate (Merk, Darmstadt, Germany) (*v*/*v*) (eluent B). A gradient elution was applied starting with 100% B for 2 min with a subsequent 14 min linear decrease to 58% B (2 min hold time) and finally a 1 min linear gradient back to 100% B (1 min hold time), resulting in a total run time of 20 min. After external calibration of glucose (≥99.5%; 1, 10, 25, 75 µg/mL; Sigma-Aldrich, Burlington, MA, USA), quantification was carried out using Excel (Microsoft Office 2021).

#### 2.3.2. LC-HRMS Analysis

For chromatographic separation, a reversed phase (RP) XBridge C18-column (150 × 2.1 mm i.d., 3.5 µm particle size—Waters, Milford, MA, USA) equipped with a pre-column (C18 4 × 3 mm i.d., Security Guard Cartridge, Phenomenex, Torrance, CA, USA) at 25 °C and a flow rate of 250 µL/min were used. As described by Bueschl et al. [43], two eluents, H_2_O with 0.1 formic acid (FA) (MS grade—Sigma-Aldrich, Burlington, MA, USA) (*v*/*v*) (eluent A) and methanol (MeOH) (Riedel de Haen, Honeywell, LC-MS grade > 99.9% purity, Seelze, Germany) with 0.1% FA (*v*/*v*) (eluent B), were applied for gradient elution. The 45 min long program started with 10% B, which was held constant for 2 min. Then, B was increased linearly to 100% within 30 min and held constant at 100% for 5 min and the column was re-equilibrated at 10% B for 8 min. During chromatographic separation, the flow rate was held constant at 250 µL/min and the column was tempered to 25 °C.

For LC-HRMS full-scan measurements, heated electrospray ionization with fast polarity switching (+/− ionization mode) was employed. The scan range was set to *m*/*z* 140–2000 and the resolving power was set to 120,000 at *m*/*z* 200. Instrument performance and chromatographic stability were monitored with a quality control mix containing 25 standard substances in MeOH: H_2_O 1:1 (*v*/*v*) containing 0.1% FA, regularly injected during the measurement sequences.

#### 2.3.3. LC-HRMS/MS Measurements

Specific inclusion lists were generated for positive and negative modes for each sample type in the spatial distribution experiment. Full scan LC-HRMS/MS chromatograms were first evaluated with MetExtract II (version 2.11.0) [44] using parameter settings listed in Appendix A. Then, each MetExtract II derived feature was placed on an inclusion list with a ±0.25 min time window around its recorded retention time. All LC-HRMS/MS measurements were separately performed in positive and negative modes with a resolving power setting of 60,000 FWHM, at *m*/*z* 200. LC-HRMS/MS spectra were recorded with stepped collision energy at 20, 45, and 70 eV.

### 2.4. Data Processing

#### 2.4.1. Conversion of LC-HRMS File Format

All Thermo Fisher LC-HRMS/MS raw files were centroided and converted into the mzXML format using ProteoWizard MSConvert Software (version 3.0.21246-1) [45].

#### 2.4.2. MetExtract II Data Processing, Metabolite Annotation of, and Integration between the Two Experiments

MetExtract II (version 2.11.0) [44] was used for data evaluation of LC-HRMS chromatograms from the culture supernatant extracts. Furthermore, chromatograms of ^12^C culture aliquots with no U-^13^C labeled internal standard served as blanks. Detailed processing parameters are available in Appendix A. The detected metabolites are represented with their accurate or presumable mass, number of carbon atoms, and type of ion species. For each metabolite, this information was queried against the Antibase 2017 database [20] and a literature-based in-house-compiled database, containing 78 additional metabolites of *T. atroviride*. To this end, the accurate or presumable mass of the noncharged metabolites in the database were compared to the exact mass of the detected unknown fungal metabolites. A mass deviation of ±5 ppm was allowed, and the number of detected carbon atoms also had to match that of the database hits.

The two experiments were combined. To this end, each feature pair was compared via *m*/*z* value and retention time. Combination parameters were set to a maximum error between the found masses in the two data sets of 5 ppm and maximum retention time shift of 0.02 min. A complete data matrix containing all found metabolites can be found in Appendix A.

#### 2.4.3. HILIC Data Evaluations

Since differences in growth under LD compared to RL and PD conditions were noticed, the remaining glucose concentration in the growth media was determined and incorporated in the normalization step during data evaluation.

The culture replicate with the highest remaining concentration of glucose was set to a weighting factor of 1. For each of the other cultures, the measured glucose concentration (in mg/kg) was divided by the maximum concentration of all experimental samples. The results of this division were then used as a weighting factor for each respective culture. Actual numbers can be found in Appendix A.

#### 2.4.4. Molecular Networking

All metabolites after MetExtract II processing, with a minimum abundance of 100,000 were compiled into inclusion lists. Then, these were used for LC-HRMS/MS measurements. The resulting combined alternating MS1 and MS/MS scans were converted to mzXML (see Section 2.4.1). Full scan data were subjected to peak picking using XCMS [46]. This included peak grouping, retention time alignment, and gap filling. MS/MS spectra were then selected and consensus spectra were created for all spectra assigned to a single metabolite. For each precursor ion, average spectra from multiple MS scans across the respective chromatographic peaks were used. Molecular networks were generated using the Feature-Based Molecular Networking workflow in the GNPS analysis environment [47]. The cosine score was set to be above 0.7 and the number of matched peaks was set to 5. Generated networks were illustrated with Cytoscape [48].

### 2.5. Statistical Data Evaluation

Statistical analysis was performed with R (version 3.5.3.) and RStudio (version 2022.07.1) with an in-house generated R script as described in [49].

The most abundant ^12^C feature per feature group (i.e., metabolite) served as the relative abundance for each fungal feature. Only metabolites that were present in at least 50% of the replicates of a light condition (LD, PD, RL) per experiment were used. 

Specific plots were generated as follows: for Venn diagrams, a metabolite was classified as being present in one group if it was present at least in 50% of the replicates.

For Volcano plot illustration, *t*-tests for each metabolite between the two compared groups were calculated. The global alpha significance threshold was set to 0.05 and a minimum fold change of 2 or a maximum of 0.5, respectively, was also required for a metabolite to be considered significantly differing between two experimental conditions. Multiple test correction was performed according to Sidak [50]. The metabolites that were found to be significantly differing for each group were then also illustrated in an UpsetR plot [51,52]. 

Multivariate statistics comprised of principal component analysis (PCA) as well as hierarchical cluster analysis (HCA). Before that, missing values in the data matrix were replaced with 0, and subsequently the data matrix was auto-scaled. Heatmap illustrations were generated using squared Euclidean distance and Ward’s linkage method.

## 3. Results

Fungal secondary metabolite production is known to be impacted by environmental cues and the culture conditions [10,19]. Based on our previous findings that 6-PP production by *T. atroviride* is stimulated upon fungal growth in darkness [40], we became interested in evaluating the influence of light on its global secondary metabolome. For the reliable and comprehensive screening of all detectable metabolites, we have used a stable isotope-assisted approach, which required the use of a synthetic medium with a fully 13C labeled carbon source. With respect to the light conditions, special focus was placed on identifying a light condition that favors metabolite production to a similar extent as cultivation in complete darkness, in order to have an alternative to the hardly feasible experimental handling under complete darkness conditions. To this end, three different light conditions were systematically compared (Figure 2, top left): (A) “Permanent darkness” (PD), in which cultivation was performed in permanent darkness and sampling was carried out under red light only. (B) “Reduced light” (RL), in which cultivation was performed in complete darkness with daily exposure to daylight for 10 min, thereby mimicking laboratory practices where the fungus typically is cultivated in darkness but is exposed to short periods of light, e.g., during growth checks or sampling. (C) “Light-dark cycle” (LD) with a 12 h dark and 12 h light period to simulate the natural day-night rhythm.

The effects of the three light regimes on the global secondary metabolome of *T. atroviride* were investigated by LC-HRMS analysis. In an additional approach, the light conditions PD and RL were compared in more detail by looking at metabolite mobility on the culture slide (Figure 2). 

As secondary metabolites are known to be involved in interspecies chemical communication [53], the spatial distribution of selected compounds was investigated with a special focus on their occurrence relative to the hyphal growth front. Substances released at the colony periphery and diffusing into the surrounding might have a function in inter-species cross-talk and mycoparasitism. Figure 2 provides an overview of the number and types of secondary metabolites detected in both approaches. Moreover, 97% of the metabolites found in the first approach (comparison of the three light regimes) were found in the second approach (analysis of spatial metabolite distribution). Metabolite annotation was based on all MetExtract II derived features and made use of accurate mass, the type of ion species, and the number of carbon atoms per formula for database search (see Section 2.4.2). Following this procedure, a total of 122 of the detected metabolites were annotated and grouped according to their biosynthetic origin (where available), or if this was not possible, according to their chemical nature (Appendix A). Peptaibols were also detected, but due to their “microheterogenity” and formation of in-source fragments, automated annotation with databases was not successful. Based on prior knowledge and by manual search, the presence of trichorzianines in the analyzed samples was confirmed. 

### 3.1. Metabolite Production under Three Different Light Regimes

LC-HRMS analysis resulted in metabolite profiles with a total of 181 features that could be assigned to 78 metabolites. The principal component analysis scores plot of the metabolite profiles revealed a variation among the *T. atroviride* cultures grown under the three light conditions tested (Figure 3). Particularly, the chemical profile produced by the fungus upon cultivation under the LD condition caused a clear separation from the samples subjected to the other two conditions (RL, PD). However, most of the metabolites were produced under all light conditions tested, as shown in Figure 3b.

These common 68 metabolites comprise well-known substances, such as 6-PP, 6-n-pentenyl-2H-pyran-2-one, and α-bergamotene. In addition, one substance was specifically produced under each of the tested light conditions. For cultivation under PD, the obtained information (*m*/*z*: 498.2822; number of carbon atoms (Xn): 22; RT: 32.61 min) indicates that the respective metabolite is a peptaibol. There was one metabolite produced only in RL treated cultures (*m*/*z*: 186.1851; Xn: 11; RT: 23.52 min). 

Two metabolites were found in both extracts derived from cultures grown in PD and LD (*m*/*z*: 318.3; Xn: 18; RT: 29.19 min and *m*/*z* 517.1182; Xn: 32; RT: 5.7 min) and five substances commonly detected in both PD and RL cultures (*m*/*z*: 150.055-499.1082; Xn: 8-16; RT: 3,98-21,96). Overall, the highest metabolite diversity was produced by *T. atroviride* upon cultivation under RL and PD conditions.

To assess whether their relative abundance is light dependent, pair-wise comparisons of the relative amounts of each metabolite were realized (Figure 4). As *T. atroviride* showed reduced growth in LD conditions compared to PD and RL, the residual amount of glucose in the cultivation medium of the cultures was determined on diluted extracts via HILIC-HRMS to estimate biomass formation and was used for the normalization of metabolite abundance.

Among the significantly differing metabolites, 24 and 22 were more abundant upon fungal growth in PD and RL compared to LD, respectively. Interestingly, 16 of those were shared between RL and PD type samples, among them 6-PP (Figure 4), respectively. When *T. atroviride* was grown in RL and PD, only six and two substances were less abundant compared to LD conditions, respectively. 

Seven metabolites were differentially abundant in the pairwise comparison between PD and RL conditions, with three being more abundant upon fungal growth in PD and four being more abundant under RL conditions.

### 3.2. Spatial Distribution of Metabolites Produced under PD and RL

In total, 1474 feature pairs allocated to 503 metabolites were detected in the different zones and under the two light conditions (RL and PD) tested. From those 503 metabolites, 122 were annotated based on full-scan data using the Antibase 2017 database [20] and an in-house curated database that was compiled of secondary metabolites from *T. atroviride* described in the literature.

Most substances belonged to terpenoids (43 of 122; 35%), the second largest group (24 of 122; 20%) were acids, and the remaining annotations corresponded to alcohols, lactones, isonitriles, ketones, oxazoles, amines, and esters (Figure 2; Appendix A).

*T. atroviride* is widely used as a biocontrol agent against phytopathogens, such as *Phytophthora cinnamomi* [54], *Botrytis cinerea* [55], and *Fusarium* spp. [56]. Therefore, mobile metabolites that are released into the environment are of special interest as they are presumably implicated in intra- or inter-species communication.

To investigate the spatial metabolite distribution along the direction of fungal growth, the culture was sliced into five 1 cm wide sections (Figure 5), each of which was analyzed separately. Notably, sections 4 and 5 (zone 2) were ahead of the hyphal growth front, and thus represent the exometabolome of the fungus.

A comparison of the number of metabolites detected in zones 1 and 2 revealed that most substances were present in all sampled parts of the culture slide and were produced under both light conditions tested (Figure 6). Twenty-five metabolites were exclusive for PD cultures. Of those, twenty substances were only present in zone 1, four in both zones, and one metabolite was exclusively detected in zone 2. Eight metabolites were specific for RL grown cultures, six of which were specific for zone 1, one substance was detected in both zones, and one metabolite was only present in zone 2. Moreover, one hundred and thirteen metabolites were exclusively found in zone 1, twenty upon cultivation in RL, six upon cultivation in PD, and eighty-seven independently of the applied light regime. Two metabolites were exclusively found in zone 2, one only upon fungal growth in RL, and one upon cultivation in PD.

To obtain a clearer picture of the spatial metabolite distribution and the influence of the two light conditions tested, a heat map was constructed using a bi-clustering approach (Figure A2 in Appendix B). The derived dendrogram was cut into four distinct clusters, each consisting of metabolites with similar expression profiles among the experimental conditions tested (Figure 7).

While for three clusters (orange, pink, gray) the detected metabolite abundances did not significantly differ between the PD and RL conditions, the abundances of the substances in the yellow cluster were significantly differing between the two light conditions (Figure 7).

The largest (orange) cluster is composed of 243 metabolites and includes the putative sesquiterpenoid trichoderiol B [21]. For both light regimes tested, the abundance of the metabolites in this cluster was highest in the first sections and decreased toward the colony margin and the non-colonized area.

The gray cluster includes harzianol J and its respective metabolites show higher abundance in the mycelial-covered zone 1, however, they continuously decreased in abundance away from the hyphal front. 

Metabolites in the pink cluster, such as melanoxadin have a similar distribution pattern as the substances of the gray cluster but higher variability between the replicates, especially when cultivated in PD. 

Metabolites assigned to the yellow cluster are more abundant in sections 2–4 upon cultivation in PD compared to RL. In addition to 6-PP, this cluster contained seven structurally similar substances, including 6-n-pentenyl-2h-pyran-2-one. Moreover, other volatiles, such as α-bergamotene are contained in this cluster (Figure 7).

To obtain more information about all detected unknown metabolites and to extend the number of compounds which can tentatively be assigned to structure classes of already annotated/identified *T. atroviride* metabolites, all with a minimum abundance of 100,000 were used for separate LC-HRMS/MS measurements. Via feature-based molecular networking, the resulting consensus spectra were clustered according to their spectral similarities [47]. One subnetwork included the MS/MS spectra of 6-PP (*m*/*z* 189.088 corresponding to the [M+Na]^+^ spectra of 6-PP, purple). Since it was identified via reference standard measurement, it provided a starting point to look for similar metabolites via the networking approach (Figure 8). With the aid of databases, annotated metabolites were found to include VOCs, such as α-curcumene, α-bergamotene, and other terpenoids, such as trichoderiol B, tricho-acorenol, and harzianol J. In addition to 6-PP, the closely related 6-n-pentenyl-2H-pyran-2-one, which only differs by an additional double bond, was detected in this network. 

## 4. Discussion

### 4.1. Metabolite Production under Three Different Light Regimes

The PCA scores plot (Figure 3a) demonstrates that the metabolite profiles produced by *T. atroviride* upon growth in RL and PD are clearly separated from those produced upon growth in LD. This corroborates previous observations that light induces distinct alterations in *T. atroviride*, visible, e.g., through conidiation [39,40,57]. It is notable that the metabolite profile-based distribution of fungal samples obtained from cultivation under PD and RL was not clearly separated, evidencing similarities in their chemical composition. As the LD condition separated more distinctly from PD than RL separates from PD, it can be concluded that LD exposure more strongly affects metabolite production by the fungus. In this context, it is also worth mentioning that UV and even regular white can be a source of oxidative stress, for example, as reviewed in [58].

Looking at the number of detected metabolites (Figure 3b), the majority of the substances were produced regardless of the light regime and only one metabolite was distinctly produced per light condition. The metabolite produced in PD only, evidently belongs to the class of peptaibols. This is in contrast to previous reports, where lowest abundances of peptaibols were found in darkness and peptaibol production was connected to conidiation and light [59]. However, the low number of metabolites being assigned to only a certain experimental condition might also be attributed to their low abundance with the respective concentrations being close to the limit of detection (LOD). Therefore, according to the definition of LOD, they might have been missed in one of the replicate samples, although they were present in the culture. 

Overall, the highest number of metabolites was detected in RL and PD cultures. This was in line with our hypothesis, which was based on previous findings, e.g., that certain secondary metabolites, such as 6-PP are most abundant in darkness [33,40]. Therefore, the highest number of metabolites may also be found under this cultivation condition. In addition, the fact that only a low number of metabolites was discovered exclusively in PD derived extracts suggests that RL can be employed for a comparable metabolic coverage in secondary metabolite analysis.

Comparing the pairwise abundance of each metabolite between the tested conditions showed a stronger difference between the three tested light conditions. A quarter of the detected metabolites were significantly more abundant under RL and PD in comparison to the LD condition (Figure 4a). The affected metabolites included 6-PP, which is in accordance with previous studies that showed that the overall amount of 6-PP was highest when *T. atroviride* was cultivated in complete darkness [33,40]. The repression of metabolite production upon growth in LD corresponds with previous findings that extensive visible light in nature is a signal for stress in fungi [42,58]. This is also in accordance with the observation that cultures in LD showed slower growth. Since the remaining amount of glucose in the cultivation media was used for metabolite abundance normalization, this effect can be disregarded. Therefore, we conclude that the lower metabolite abundance upon growth in LD can be traced back to the general repression of secondary metabolite biosynthesis under this light condition.

In summary, we found that the majority of metabolites are produced by the fungus regardless of the used light condition, while only 13% (10/78) of the detected substances were selectively influenced by the applied light regime. Most of them were produced during low light, i.e., PD and RL conditions. Additionally, about 25% of the metabolites were significantly more abundant when *T. atroviride* was cultivated under low light. From those significantly enhanced metabolites, 16 were more abundant in both RL and PD compared to LD. Therefore, we could show that cultivation of *T. atroviride* under RL results in a similar metabolome coverage and metabolite profile as for growth in PD. Based on these findings, these two conditions were investigated in more detail by comparing the spatial distribution of the produced metabolites on the culture slides.

### 4.2. Spatial Distribution of Metabolites Produced under PD and RL

Seventy six out of the 78 metabolites, which are detected when culturing *T. atroviride* under the three tested light conditions, are also found in the approach examining the spatial distribution of fungal metabolites along the culture grown under PD and RL. The considerably lower number of metabolites detectable in the cultures grown under the three tested light conditions compared to the second approach can largely be explained by the lower degree of ^13^C enrichment. The lower ^13^C enrichment resulted in broader isotope patterns of the corresponding ^13^C isotopologues, and thus lower signal intensities for the labeled metabolite ions. This led to a lower number of metabolites fulfilling the strict filtering criteria of the MetExtract II program. The presence of numerous compounds, which had been missed by automated data processing in the first experiment, was successfully verified by manual inspection of LC-HRMS data. Most of them were not found with MetExtract II since their signals were below the intensity cut-off used for feature finding. For consistency in the processing workflow between the two experiments, we decided to use the same strict processing parameters for both experiments (Appendix A).

Regarding the chemical structure of the detected metabolites, the largest group (43/122; 35%) consists of terpenoids. This is plausible, as terpenoids are the biggest class of secondary metabolites described for *Trichoderma* spp. to date, and have various biological activities [60,61].

The second largest group putatively are acids (24/122; 20%), such as trichodermic acid A. Interestingly, this metabolite has not yet been reported to be produced by this *Trichoderma* species [62].

In a former study, Klitgaard et al. have successfully used a feature-based molecular networking (FBMN) approach in combination with stable isotopic labeling to elucidate biosynthetic routes of small fungal peptides [63]. Here, we used a similar approach to extend the number of compounds which can tentatively be assigned to structure classes of already annotated/identified *T. atroviride* metabolites based on the similarity of their product ion spectra [47]. A special focus was placed on the network cluster that included 6-PP (Figure 8), since this pyrone is assumed to play a central role in the interaction of *T. atroviride* with its environment. It is interesting to note that the additional metabolites annotated in the cluster of 6-PP were not annotated as related pyrones but belonged to the class of terpenoids (e.g., tricho-acorenol [64] or α-curcumene [65]). The assignment of terpenoids to the network cluster linked to 6-PP can be explained by the pentyl side chain of 6-PP and its alkyl moiety putatively obtained after cleavage of the lactone ring and loss of CO_2_, resulting in an MS/MS fragment pattern similar to those of terpenoids. 

Next, all detected metabolites were grouped according to their occurrence in the section of the culture slide that was covered by mycelium (zone 1) and the region in front of the hyphal growth front (zone 2) as well as the tested light condition (Figure 7). The study design on solid media was chosen in order to be able to give a rough estimation of the location of the detected metabolites on the culture slides. 

Different types of secondary metabolites serve different biological functions and are produced as families of related substances, often during a specific stage of morphological differentiation [66]. Metabolites involved in mycoparasitism support *T. atroviride* in killing its fungal prey in combination with hydrolytic enzymes that can lyse the cell wall of the prey fungus [67]. Chemical crosstalk can already occur before physical contact between *Trichoderma* and its interaction partner is established. Metabolites that are released by the fungus and can be found ahead of the hyphal growth front have been termed the exometabolome and can be assumed to exhibit a role in the interaction with the environment. In contrast, metabolites remaining inside the producing fungus were termed the endometabolome. Many constituents of the endometabolome can be assumed to serve different functions than those released to the environment [68,69].

In our study, zone 1 contained the highest number of metabolites. This was expected, since from these culture sections, intracellular and extracellular metabolites were extracted. The type and profiles of these metabolites only show minor differences between cultures grown in RL or PD, which is in accordance with the first experiment. Nevertheless, 20 metabolites were only found in zone 1 after cultivation in PD in comparison to six compounds, which were only found in zone 1 in RL cultures. Zone 2 contains only metabolites that were released into the medium. Overall, two of the detected biochemical constituents were exclusively detected in the sections ahead of the hyphal front (e.g., the exometabolome), one in RL and one in PD. With the used databases, however, we were not able to annotate the two metabolites exclusively found in zone 2. Surprisingly, many metabolites are also present in zone 2, and thus are mobile enough to potentially be involved in the interaction of *T. atroviride* with its environment.

To gain more insight into the spatial abundance profiles of the detected metabolites and the dependence of their production on the used light conditions, all compounds were clustered according to their spatial distribution profiles for each of the two tested light regimes (Figure 8). With 243 compounds, the largest cluster (orange) shows steadily decreasing abundances from the inoculation point along the direction of growth. Among those, 18 metabolites consisted of 15 carbon atoms, making them strong candidates to be sesquiterpenes [60,70]. The abundance profiles in this cluster suggest that the corresponding metabolites were produced by *T. atroviride* constantly throughout the cultivation period and migrate freely through the medium, which includes diffusion in front of the hyphal growth front. Interestingly, on average the metabolites contained in the orange cluster are slightly more abundant under RL conditions, but this difference is not statistically significant.

A recent study investigated the effect of the circadian clock on the volatile and diffusible secondary metabolome of *T. atroviride* strain IMI 206040. A supposed secondary metabolite with a *m*/*z* value of 371.03 was reported, presumably fitting to a compound that we also found (MET463; putative molecular weight of neutral molecule 371.1829; number of carbon atoms 16, positive ionization mode) within the orange cluster [71]. 

The spatial distribution profile of metabolites contained in the gray and pink clusters suggests that those are rather produced at the hyphal growth front. Their *m*/*z* values cover the large range from 143.0341 to 1122.6511. Seven metabolites could be annotated, five of which belong to the class of terpenoids and two oxazoles. 

In contrast to the three other clusters, abundance levels of the metabolites contained in the yellow cluster are significantly affected by the applied light regime. When grown in PD, metabolite concentrations are always higher compared to RL. They strongly increase along the fungal growth direction and reach a maximum in section 3 at the colony growth front, and then rapidly decrease thereafter. In contrast, under RL conditions, concentrations are constantly lower and changed little in the direction of growth. Interestingly, 6-PP also belongs to this cluster. These data are in good agreement with the above-mentioned study, which investigated the effect of the circadian clock on secondary metabolite production including 6-PP in *T. atroviride* [71]. Higher production of 6-PP in PD has also been found in another recent study [42].

In addition to 6-PP, the closely related 6-n-pentenyl-2H-pyran-2-one was detected along with a number of volatile terpenoids including α-bergamotene, trichoderiol, harzianol, and tricho-acorenol (also found in the molecular network displayed in Figure 8). While 6-PP serves multiple functions in *T. atroviride* and sesquiterpenes, such as tricho-acorenol, have been found to promote own survival or growth during mycoparasitic interactions in fungi [32,33,72], the role of the other terpenoids that were detected in the molecular network remains to be clarified. 

Through the clustering approach, we found that the production of 6-PP and 84 other metabolites was significantly enhanced upon growth of *T. atroviride* in complete darkness compared to the reduced light exposure regime. However, for the majority of the detected metabolites, no significant effect of the light condition could be found. 

The results of our study are in good agreement with previous reports showing that the highest number and amounts of secondary metabolites are produced by *T. atroviride* upon growth in the presence of low light (RD, PD). Investigating abundance, along with the spatial distribution of the produced metabolites, revealed that 85 (16%) were significantly more abundant upon cultivation of the fungus in permanent darkness. Moreover, 418/503 (83%) of the found metabolites were not significantly more abundant and the vast majority can be found in both light conditions. These results reveal that reduced light exposure is a suitable and easily manageable condition for fungal cultivation in secondary metabolite studies of *T. atroviride.* We also investigated secreted metabolites at the front of the colony periphery that serve potential signaling and communication roles. From the detected 503 metabolites, 316 (63%), including the known chemical signaling molecule 6-PP, were present in the analyzed sections ahead of the hyphal growth front. We acknowledge that in agricultural settings on the field, other influences that we have controlled with our setup (pH, nitrogen availability, light source) may also strongly influence secondary metabolite production. Nevertheless, the results obtained in this study provide extensive information on over 500 *T. atroviride* secondary metabolites that will be of high usefulness for the annotation and discussion of the metabolome of this mycoparasite in future investigations.

## Figures and Tables

**Figure 1 jof-09-00785-f001:**
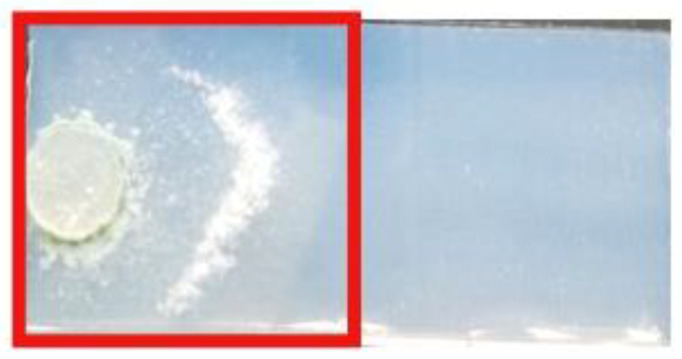
*T. atroviride* colony grown on a slide covered with minimal medium. The photo was taken just before sampling of the red-labeled area excluding the inoculation plug.

**Figure 2 jof-09-00785-f002:**
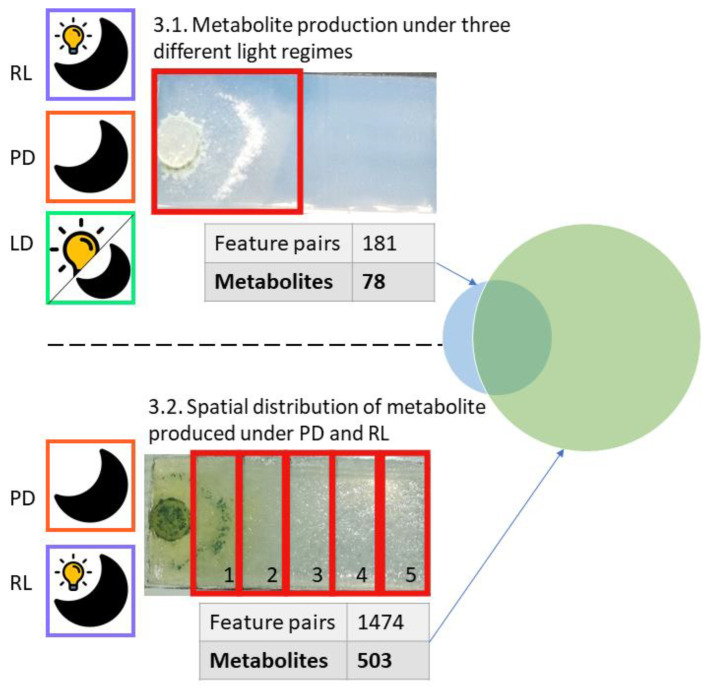
Overview of the performed comparisons. The found metabolites in both experiments were compared as illustrated by the Venn diagram. Of these, 122 were annotated based on Antibase 2017 [20] and a literature-based in-house curated *Trichoderma atroviride* database.

**Figure 3 jof-09-00785-f003:**
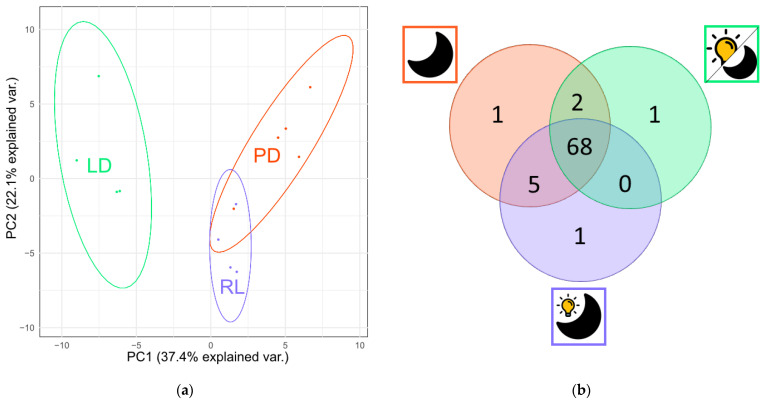
(**a**) PCA scores plot comparing the three light conditions. (**b**) Venn diagram showing the number of detected metabolites in at least 50% of the replicates.

**Figure 4 jof-09-00785-f004:**
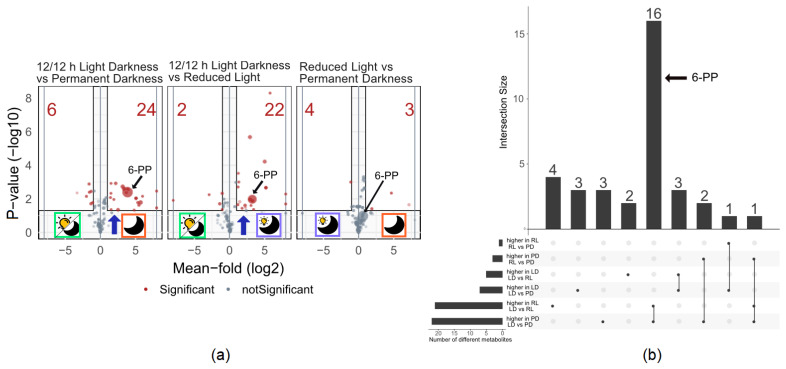
(**a**) Volcano plots showing the pairwise comparison of the conditions tested. The size of the dots indicates the mean abundance in the more abundant group. The red numbers indicate the number of metabolites being significantly more abundant in the respective group in comparison to the other. (**b**) UpsetR plot displays the significantly differing metabolites illustrated on the left and middle volcano plots of (**a**), respectively. 22 or 24 metabolites were significantly enhanced in RL compared to LD or PD compared to LD respectively. 16 of those were shared between RL and PD.

**Figure 5 jof-09-00785-f005:**
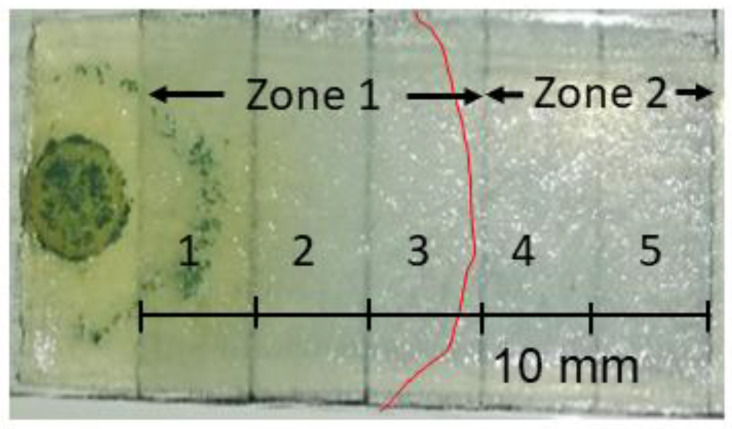
Cultivation slide with the five separately harvested and extracted sections. The red line indicates the hyphal growth front. At the time of sampling, sections 1–3 were covered by fungal mycelium (zone 1), while sections 4 and 5 were not (zone 2).

**Figure 6 jof-09-00785-f006:**
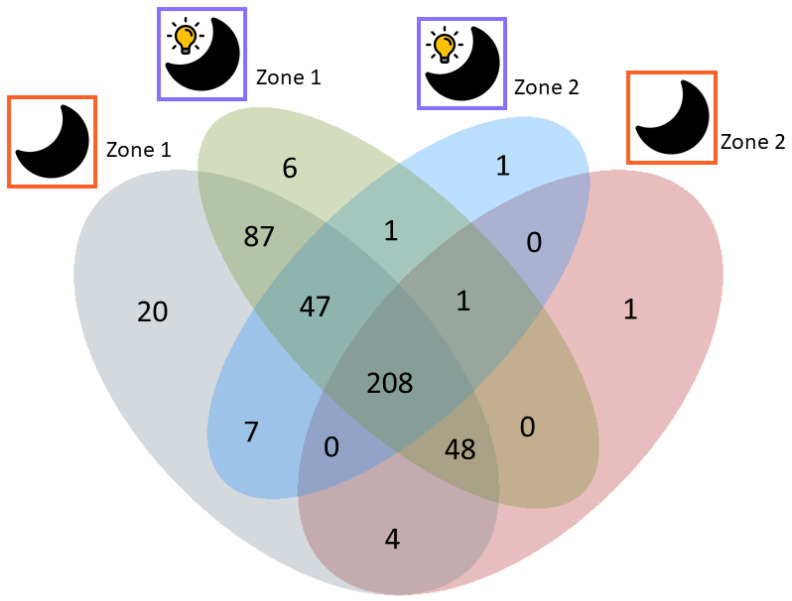
Venn diagram displaying the number of metabolites found only in parts of the slide that belong to the endometabolome and exometabolome (zones 1–3) or only the exometabolome (zones 4–5). Only metabolites found in at least 50% of the replicates per tested condition (zone 1 RL, zone 2 RL, zone 1 PD, zone 2 PD) are displayed.

**Figure 7 jof-09-00785-f007:**
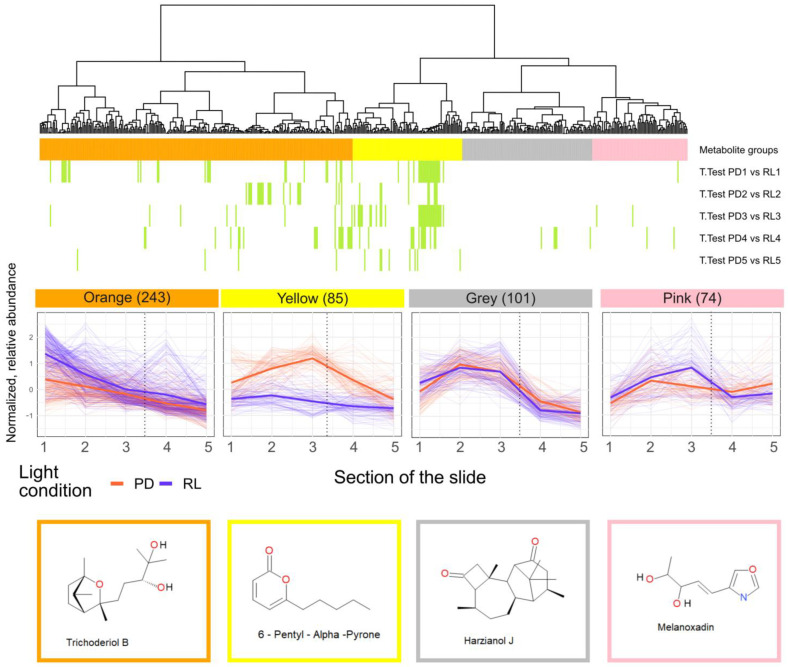
Heat map clustering results of the spatial metabolite distribution under RL and PD conditions. Hierarchical bi-clustering resulted in four distinct spatial profiles, with regard to their metabolite abundance on the five zones. The dotted line marks the border between the sections covered (zones 1–3) or uncovered (zones 4 and 5) by the fungal colony. Green bands visualize metabolites significantly differing between the two tested light conditions for each section of the slide. Each red (PD) and turquoise (RL) line in the relative abundance illustration represents the average abundance of one metabolite in each section, while the bold line for each cultivation condition illustrates the average of all metabolite averages in this condition.

**Figure 8 jof-09-00785-f008:**
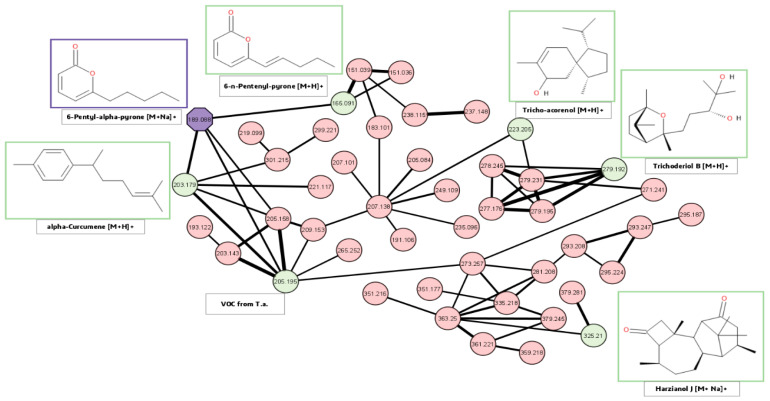
Feature-based molecular network that includes the precursor ions directly linked to the known sodium adduct of 6-PP (*m*/*z* 189.088; highlighted in violet). Each node in the network shows the *m*/*z* value of the corresponding precursor ion. The cut-off for the cosine score was set to 0.7. The edge thickness is proportional to the cosine score between the corresponding two nodes. Six additional precursor masses of metabolites were putatively annotated, using the underlying stable isotopic labeling information (number of detected carbon atoms and calculated sum formula) with a database comparison (Antibase 2017 [10]). Annotated metabolites are highlighted in green. For the *m*/*z* value of 205.195, more than one annotation was possible.

## Data Availability

Not applicable.

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
