# Peer review of "Light-Induced Changes in Secondary Metabolite Production of Trichoderma atroviride"

_jof, 2023, doi:10.3390/jof9080785_

Round 1

Reviewer 1 Report

The authors investigated the effects of light condition changes (dark, light, or dark light cycle ) on metabolomics shifts of Trichoderma atroviride. An interesting work, which would maximize secondary metabolite production of fungi in agriculture or industry.

This manuscript was well organized and furtherly can be accepted, but I have some minor concerns which should be addressed first.

1.      In section of introduction, the role of bioactive Trichoderma metabolites on plant pathogens should be give more details.

2.      Line 109, why chose minimal medium? It should be explained.

3.      Line 393-396, metabolites were annotated by Antibase and in house database. Annotation is very important in metabolomics study, so the detailed annotation process should be mentioned in method.

4.      6-PP, 6-pentyl-α-pyrone is easy to volatilize, maybe GCMS is suitable for its detection, as well as other volatiles. Here LCMS could detect 6pp? 

Reviewer 2 Report

Fungal secondary metabolism is an interesting filed which is studied by many researchers. Light is an important signal that fungi perceive through their light receptors and controls their development and production of secondary metabolites. Although the light signalling has been studied in fungi well, it was not studied substantial enough with respect to secondary metabolite production. There are only very few studies looking at production of secondary metabolites in dark and light. In this paper entitled " Light-induced changes in secondary metabolite production of Trichoderma atroviride, the authors determined total production of secondary metabolites under three different light regimes, continuous darkness (PD), reduced light exposure (RL) and alternating light and dark exposure (LD). They took challenging approach which focused on untargeted metabolomics using LC-MS. Interestingly Trichoderma produced similar metabolites both under continuous dark and light conditions. This indicates that the fungus turn on metabolic pathways related with secondary metabolism better under same conditions. However, alteration from light to dark have a negative impact on the diversity and abundance of the secondary metabolites in this fungus. The authors interestingly detected a total of 500 metabolites in this approach. A great proportion of the metabolites were detected under low light illumination, suggesting that neither light nor darkness is good for secondary metabolite production in this fungus. Finally authors concluded that the low light condition could be the best condition to maximise production of secondary metabolites in this fungus. Overall, the experiments performed and underlying data supports the conclusions that the authors have made. Only little suggestion is that they might discuss the light as a stressor for the fungus, It is interesting that switch from light to dark or dark to light generate stress for the fungus so that it might result in inhibition of secondary metabolites. However, normally stress conditions allow the fungus to produce secondary metabolites. Maybe light inhibition of metabolic pathways can be discussed in discussion in more detailed or light as a stress signal can also be mentioned.

Reviewer 3 Report

The manuscript "Light-induced changes in secondary metabolite production of Trichoderma atroviride, by K. Missbach et al., describes a technically very elaborated work, in which its main merit is the effort to decipher the metabolome of Trichoderma in very specific culture conditions. Such conditions were chosen apparently for its resemblance with environmental circumstances in which it can interact at a distance with other microorganisms, to which presumably it can control with the help of a complex chemical arsenal. The article focuses however on the effect of light, choosing the comparison between darkness and a 12 h light / 12 h dark regimes, and in a very arbitrary way (in advance of experimental practicality), darkness with alternating 10 min of light every 12 h. The major conclusions is that the 12h light/ 12h dark regime affects negatively to the the diversity of the produced metabolites, but there is scarce effect of short illuminations compared to darkness. This is not surprising, considering that it is basically a dark illumination, with very short sporadical illuminations with environmental light, presumably not as strong as the illumination in the chamber.
The work is interesting, although the way in which the results are presented is addressed to metabolomics experts rather than to researchers interested in secondary metabolism and its regulation. The latter may have some difficulties in interpreting the data in the way they are presented, and the methodology is sophisticated and difficult to follow for non-experts. I am rather of the second class, and for me as a reviewer the methods are excessively specialized,  I can only hope that the methodology is correct and that it has been applied correctly. I cannot go into this aspect of the work, and I leave its evaluation to reviewers with more knowledge of these techniques.

The work may be worth of publication, but requires revision. I find some drawbacks, which I describe below, that in my opinion should be taken take into consideration before publication:

1. It seems there is a considerable mess with the types of secondary metabolites produced by Trichoderma, and this manuscript does not help to clarify it. Many appear to be very small molecules, often volatile, whose chemical classification does not fit the traditional classification in terpenoids, polyketides, non-ribosomal peptides, etc, (see as a recent comprehensive review doi.org/10.3390/encyclopedia2010001). The mess is appreciated in the paragraphs starting from Line 62 in the introduction, which makes the provided information inconsistent. The described list starts with large chemical families (polyketides, terpenes), then goes to volatile compounds (volatile is not a type of chemical compound, but a physical quality) and it ends with very specific compounds, instead of mentioning about other large families, as possible short non-ribosomal peptides (are there any?) or products from combined PKSs-NRPs, as found in other fungi.
Also in the same direction:
- It starts saying "Secondary metabolites known to be produced, but in the next paragraphs talks of "unknown" putative polyketides.
- The classification list of the 122 annotated metabolites that appears in Figure 1 makes no sense. Alcohols, acids, ketones... are very general chemical terms, while oxazole are a very specific family of molecules. Terpenoids are conceptually different, and they are actually too diverse. Some terpenoids are very small, others very large, it includes alcohols, or acids... The chemical categories of this list are not comparable, and it makes it rather useless.

2. In the way in which the experiments are designed, apart from the problems arising from the sensitivity of the detecting method mentioned in lines 485-490, many metabolites are possibly not detected because the regulatory conditions for their production are not present. Very probably there are more decisive signals for their production, such as nitrogen availability, or pH. It must be noted also that the effect of light can change depending on the conditions used. Something should be mentioned at this respect. From the biotechnological point of view, the conditions used are far form those typically used for biotechniolgical purposes, normally in submerged cultures (frequently in bioreactors). I understand it is not the objective of the work, what it must be clearly reflected in the manuscript.

3. About the experimental design: the strip separation at different distances discriminates metabolites according to their degree of diffusion, but as the authors themselves repeatedly point out, in zone 1 it does not distinguish between metabolites within the hyphae or secreted into the medium. Again, the ideal would have been a submerged culture, in which mycelium and culture medium could be easily separated by filtration. I understand the reasons of the authors, but whhy submerged culture in liquid medium was not used should be let more clear.

4. Diffusion is "blind". It must be kept in mind that diffusion must occur in any direction, and if metabolites are produced at the tips of the hyphae (band 3) they not only diffuse forward but also backward. However diffusion is a physical property that should correlate with chemical parameters, especially size, and may be also shape and polarity. Have the authors tried to correlate the presence of the chemicals in strip 5 with chemical properties?

5. I disagree with the use of the glucose data. The initial medium has 10 g/l glucose, and the amounts they measure (shown in Supplementary Table 2) are about 60 µg/ml. That is 0.06 g/l.  It means that in all cases there is only residual glucose. I am not surprised, since 10 g/l is rather a low glucose amount. But in practical terms, all the samples have consumed practically all the glucose. In my opinion, no corrections are needed.

6. The discussion is too long and it would need in my opinion some rewriting, removing repetitive text with the results, and including however some aspects that have been overlooked, mentioned in the above comments.

Specific comments:

Figure numbering in the text correspond to the previous figure (Figure 2 is Figure 1, Figure 3 is Figure 2, and so on). It seems a figure was removed lately and Figure numbering was not updated. This has not helped in the reading.

Line 52: what do you mean with dynamic taxonomy? That it eventually changes as criteria of researchers change?

Line 56 This reference of Demain is from 1987. There is not a more recent reference to be mentioned here?

Lines 93-94. Production of conidia is a developmental phenomenon, rather than physiological or biochemical

Lines 114-120: I was surprised that the minimal medium has only 0,05 g per liter of ammonium sulphate as nitrogen source. In my experience, these are very strict nitrogen-limiting conditions. Is that correct? Please, note that as mentioned above, nitrogen availability is a key signal for the production of secondary metabolites in other fungi.

Line 140: there are no pictures in Supplementary table 1. The legend of the table is "Parameter settings for data processing using MetExtract II". It seems tables 1 and 2 have interchanged.

Line 141: there is no reason for not having pictures of the dark cultures. Parallel samples may be incubated and used for photography at different growth times and be discarded afterwards.

In Figure 1 it is unclear the relation between the 78 and 503 metabolites and the 122 of the right scheme. Legend of Figure 1 is very insufficient to understand.

Lines 328 - 330. Note that diffusion conditions need not be the same in agar as in the environment where it is found in nature.

The text needs revising. E.g.:
Line 301 correct atrovirde
Line 306, correct completedark
Line 455 correct mesurements

Round 2

Reviewer 3 Report

The authors have adequately answered all the points I have raised. I still find some inconsistencies in the treatment of secondary metabolite, like putting for example in the same type of categories terpenoids and acids, when both things have nothing to do: actually there are acidic terpenoids, (e.g. gibberellic acid, torularhodin...). This is not an importatn concern, however. There are also improvable aspects, or at least debatable, in the way the experiments have been designed, but it is up to the authors to decide on their experimental approaches, and I agree that they have done a very interesting work, technically meritorious, and worth of publication.